# Disentangling Abstraction from Statistical Pattern Matching in Human and Machine Learning

**Sreejan Kumar** [1]*, **Ishita Dasgupta**[2], **Nathaniel D. Daw**[1,3], **Jonathan. D. Cohen**[1,3], **Thomas L. Griffiths**[3,4]

**1** Neuroscience Institute, Princeton University, Princeton, New Jersey, United States of America, **2** Google DeepMind, London, United Kingdom, **3** Department of Psychology, Princeton University, Princeton, New Jersey, United States of America, **4** Department of Computer Science, Princeton University, Princeton, New Jersey, United States of America

* sreejank@princeton.edu

**Data Availability Statement:** Data and code to produce the analyses are in the GithHub repository named Abstract_Neural_Metamers (which can be

## Abstract

The ability to acquire abstract knowledge is a hallmark of human intelligence and is believed by many to be one of the core differences between humans and neural network models. Agents can be endowed with an inductive bias towards abstraction through meta-learning, where they are trained on a distribution of tasks that share some abstract structure that can be learned and applied. However, because neural networks are hard to interpret, it can be difficult to tell whether agents have learned the underlying abstraction, or alternatively statistical patterns that are characteristic of that abstraction. In this work, we compare the performance of humans and agents in a meta-reinforcement learning paradigm in which tasks are generated from abstract rules. We define a novel methodology for building "task metamers" that closely match the statistics of the abstract tasks but use a different underlying generative process, and evaluate performance on both abstract and metamer tasks. We find that humans perform better at abstract tasks than metamer tasks whereas common neural network architectures typically perform worse on the abstract tasks than the matched metamers. This work provides a foundation for characterizing differences between humans and machine learning that can be used in future work towards developing machines with more human-like behavior.

## Author summary

There has been a recent explosion of progress in artificial intelligence models, in the form of neural network models. As these models achieve human-level performance in a variety of task domains, one may ask what exactly is the difference between these models and human intelligence. Many researchers have hypothesized that neural networks often learn to solve problems through simple pattern matching while humans can often understand a problem's underlying abstract concepts or causal mechanisms and solve it using reasoning. Because it is difficult to tell which of these two strategies is being employed in

found in https://github.com/sreejank/Abstract_ Neural_Metamers).

**Funding:** This work was supported by the National Institute of Health (NIH T32MH065214 to SK), the U.S. Army Research Office (W911NF-16-1-0474 to ND), the Darpa L2M Program (to TLG), and the John Templeton Foundation (to JDC). The opinions expressed in this publication are those of the authors and do not necessarily reflect the views of the funders. The funders had no role in study design, data collection and analysis, decision to publish, or preparation of the manuscript.

**Competing interests:** The authors have declared that no competing interests exist.

problem solving, this work develops a method to disentangle the two from a human's or neural network model's behavior. The findings confirm that humans typically use abstraction to solve problems whereas neural networks typically use pattern matching instead.

## Introduction

A key component of human intelligence is the ability to acquire, represent, and use abstract knowledge that efficiently captures the essential structure of the world, and can be used to generalize beyond the specific learning context [1]. The human capacity for abstraction has been studied from the earliest days of cognitive science. Hull [2] posited that humans extract "generalizing abstractions" that form concepts from commonalities across multiple experiences and apply them to future experiences. Bruner, Goodnow, and Austin [3] further show how these abstractions can emerge through decision strategies that manage objectives such as information gain, cognitive strain, and trade-off of risk and reward. The acquisition of abstract knowledge in the context of visual processing has roots in Gestalt psychology, which investigates how humans can extract a "whole that is more than the sum of its parts" [4]. Today, the acquisition of such generalizing abstractions has been shown to underpin characteristically human capabilities, such as extracting relational information between objects [5] to permit generalization to novel scenes and locations [6,7] guiding efficient exploration and learning [8], supporting rapid generalization of learned motor skills [9], and facilitating communication and coordination in cooperative tasks [10].

The ability to learn rapidly from small amounts of experience and then to generalize systematically beyond the learning context, as facilitated by abstraction, has also been proposed as one of the most salient differences between humans and deep neural networks [11,12,13,14]. Inspired by this, artificial intelligence (AI) and cognitive science researchers have created tasks and datasets that contain extensive abstract structure [15,16,17,18]. These have been difficult for otherwise state-of-the-art neural networks to master without specifically building the task's abstract structure into the learner, often with symbolic machinery [19]. One approach to bestowing abstractions on a neural network–without expressly building it in with symbolic machinery–is via meta-learning [20,21]. In many meta-learning paradigms that aim to produce human-like behavior [22,23,24,25,26], abstract rules are used to generate a distribution of tasks. The agent is then trained on these tasks, with the intent that the agent acquires the abstractions underlying this task distribution as an inductive bias. This is tested by examining performance on held-out tasks from this distribution, in which the acquired inductive bias is expected to facilitate fast learning. This approach has demonstrated that relatively simple neural-network models can acquire sophisticated human-like abilities including causal reasoning [22], compositional generalization [23], linguistic structure [24], and theory of mind [25].

In evaluating these models we encounter an issue that is widespread in machine learning research with deep neural networks: that neural networks are not easily interpretable. Simply examining test performance does not give full insight into what internal representations the model has learned and uses to solve a task. For example, CNNs trained on ImageNet can achieve good performance despite having encoded a set of features that differs considerably from those humans use for the same task [27]. This raises the question of whether neural networks trained on abstract task distributions actually acquire abstract knowledge, or if they learn other statistical features that might correlate with or be downstream consequences of these abstract rules.

We hypothesize that even on distributions of tasks generated from abstract rules, neural-network meta-learners do not necessarily internalize the abstract structure of those rules, despite performing well. Rather, they may learn the (potentially more complex) *statistical* structure of the stimuli and corresponding responses that are associated with the rules, without encoding the abstract rules themselves. This is in contrast with humans, who are posited (innately and/or through lifelong learning) to represent, identify, and use abstract rules in such settings [28,29,30]. It is often difficult to distinguish whether a human or artificial agent is making direct use of abstract structure versus the statistics associated with structure, because the difference between the two can be subtle and difficult to operationalize. That said, it is generally assumed that abstract structure reflects an underlying generative process that is simpler and lower-dimensional than a description of the statistics of the representations that such processes produce [1], and that inferring such structure permits more reliable generalization to novel instances than use of the associated statistical structure.

In an effort to make this subtle difference more concrete, and to test these hypotheses concerning the distinction between these two forms of inference, we present a novel methodology for distinguishing between the use of abstractions versus statistics through the use of "task metamers." We borrow the concept of a metamer from the field of human vision, in which metamers are pairs of color stimuli that have different underlying spectral power distributions but are nevertheless perceived as the same by the human eye [31]. A similar idea has also been studied using artificial neural networks in the auditory domain [32]. Here, we develop *task* metamers–that is, tasks with different underlying generative processes but that share highly similar statistical properties. Our work builds on a larger effort to construct tasks or task manipulations informed by cognitive psychology that interrogate neural networks' abilities to exhibit properties of human cognition [33,34].

In particular, we define task distributions with simple underlying abstract rules. We then develop corresponding task metamers for each set of abstract rules, which are task distributions that are generated by statistical procedures to match the abstract task distributions but are not generated by applying the rules. Our results show that humans, on average, perform better on tasks generated from the rules (*abstract tasks*) than on the statistically matched set (*metamer tasks*), whereas neural-network meta-learners, on average, perform comparably across the two and in fact do slightly better on the metamer tasks. Furthermore, we show that even when comparing performance within each set of tasks, humans perform significantly better than agents on abstract tasks, while neural networks perform significantly better than humans on metamer tasks. This double dissociation is consistent with the hypothesis that humans use underlying abstract structure to perform these tasks, whereas widely-used neural networks learn statistical patterns that are correlated with such structure. The task metamer methodology we present in this paper lets us disentangle the very subtle difference between learning abstractions versus statistical patterns. Our experiments utilizing this method provide preliminary evidence that humans are predisposed toward abstract structure, while many artificial neural agents are predisposed to learning statistical patterns.

## Results

### A Meta-Reinforcement learning task with abstract task distributions

To directly compare the performance of humans and machine-learning systems with respect to different task structures, we used a simple tile-revealing task (**Fig 1A**). Humans and machine-learning systems (henceforth referred to as *agents*) are shown a 7 × 7 grid of tiles all of which are initially white except for one which is red. Clicking white tiles reveals them to be either red or blue. The goal of the task is to reveal all of the red tiles while revealing as few blue

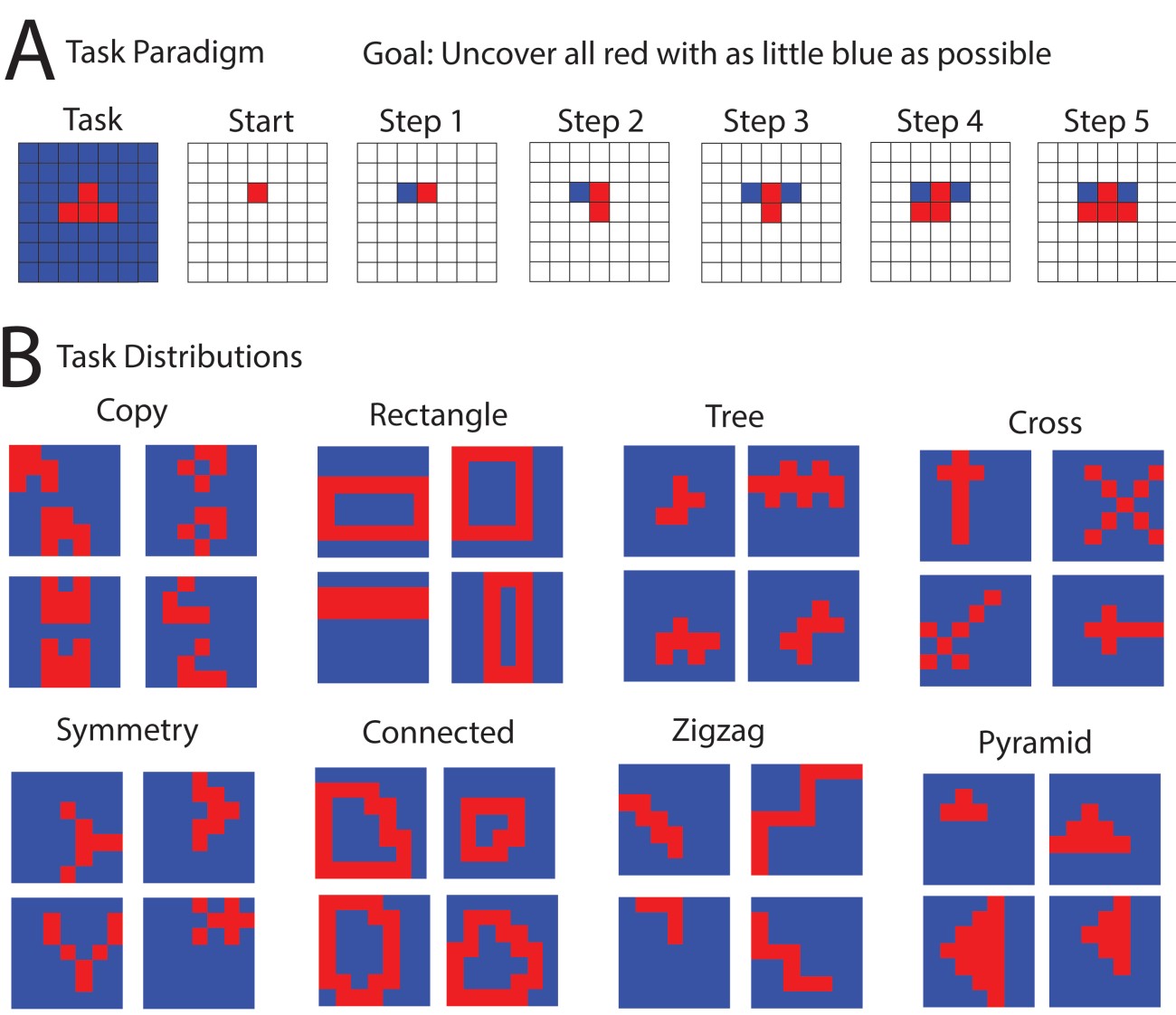

**Fig 1. Tile-revealing task.** (A) An agent sequentially reveals tiles to uncover a shape on a 2D grid. Each task is defined by the underlying board. (B) The underlying boards are sampled from a specific abstraction, which defines a distribution of boards based on an abstract rule. More examples of boards from each distribution can be found in **S1 Fig**.

ones as possible. The task ends when all the red tiles are revealed. There is a reward for each red tile revealed, and a penalty for every blue tile revealed. One grid with a fixed configuration of red tiles defines a single task. The sense in which a grid corresponds to a task is that its configuration of red tiles uniquely specifies the action to reward mapping, and therefore the task being performed by a human or agent. A distribution of tasks (or *boards*, by analogy to board games) is generated by creating boards with various configurations of red tiles determined according to a particular set of rules that go along with a particular abstraction (abstract task distribution). These distributions are shown in **Fig 1B**. A similar task was developed by Markant and Gureckis [35] to study information seeking in people, in which the underlying boards contained rectangular shapes and letters. Our work builds on this by using a suite of various abstractions, which each generate a distribution of boards, in order to evaluate whether and

how human behavior on such tasks differs from that of artificial neural networks trained on the same tasks.

We tested eight abstractions for generating boards that produce recognizable patterns (**Fig 1B**). Four of these (copy, symmetry, rectangle, and connected) were inspired by predicates proposed in Marvin Minsky and Seymour Papert's book *Perceptrons* [36], designed to be difficult for simple perceptron models to learn. Another four (tree, pyramid, cross, and zigzag) were based on general abstract structures. Examples are shown in **Fig 1B**, with detailed descriptions of how boards are generated for each abstraction provided in **S1 Table**. These rules can be considered abstract in that they are common across all boards within the same distribution, are based on concepts not necessarily tied to individual boards or the two-dimensional grid domain in general, and are relatively simple to describe.

## Generating metamer task distributions from abstract task distributions

Previous approaches to evaluating whether machine-learning systems can extract abstract structure [37,38] have relied on examining average performance on held-out examples from structured task distributions. This approach may not reliably distinguish whether a system has truly internalized the underlying abstraction or whether it has learned statistical patterns in the stimuli that are correlated with the rules, allowing it to perform the task without having specifically encoded the rules themselves. Although the difference between these two behaviors may be subtle, our goal is to produce a method that is sensitive enough to systematically distinguish them.

To directly examine whether abstract structure is a factor in how humans and meta-learning agents perform this task, we constructed a set of control task distributions that were similar in their statistical properties to the rule-generated distributions, but did not share the same abstract structure. To do so, we trained a fully connected neural network to learn the conditional distribution of each tile given all the other tiles. We did this by having the network predict the value of a missing tile given the other tiles (**Fig 2A**). We trained a different network for each rule, each of which typically achieved over 95% training accuracy (see **S1 Fig**). This is a commonly used strategy for training neural networks in both language [39] and vision [40] that encourages neural networks to learn a representation of the data that allows them to interpolate missing parts of the data. This representation has its own biases that we would like to distill into control task distribution.

We then sampled boards from the network's learned conditionals with Gibbs sampling [41]. Gibbs sampling is a common Markov chain Monte Carlo algorithm that is used to generate samples from distributions with very large support. It provides a convenient way for us to sample from the biases that the network has learned once exposed to the boards' various conditional distributions. Specifically, we started with a randomly generated board. We then masked out a single tile, determined the probability of the masked tile being red versus blue as judged by the network and turned the tile red according to this probability. This procedure was then repeated for all other tiles in the board, to give a single Gibbs "sweep." We completed 20 such sweeps. Each sequence of decisions made by the network implements a Markov chain. The stationary distribution of this chain corresponds to the distribution encoded by the network making the decision, which was trained on the conditional distributions of the boards. Accordingly, this procedure yielded samples from a "metamer" distribution of boards that matched the statistics of the abstract board, but were not generated directly by the abstract rule. We confirmed this by analyzing the first-, second-, and third-order statistics of each metamer and determining the extent to which they were statistically indistinguishable from the corresponding abstract distribution (see **Fig 2C**). These statistics were chosen as simple

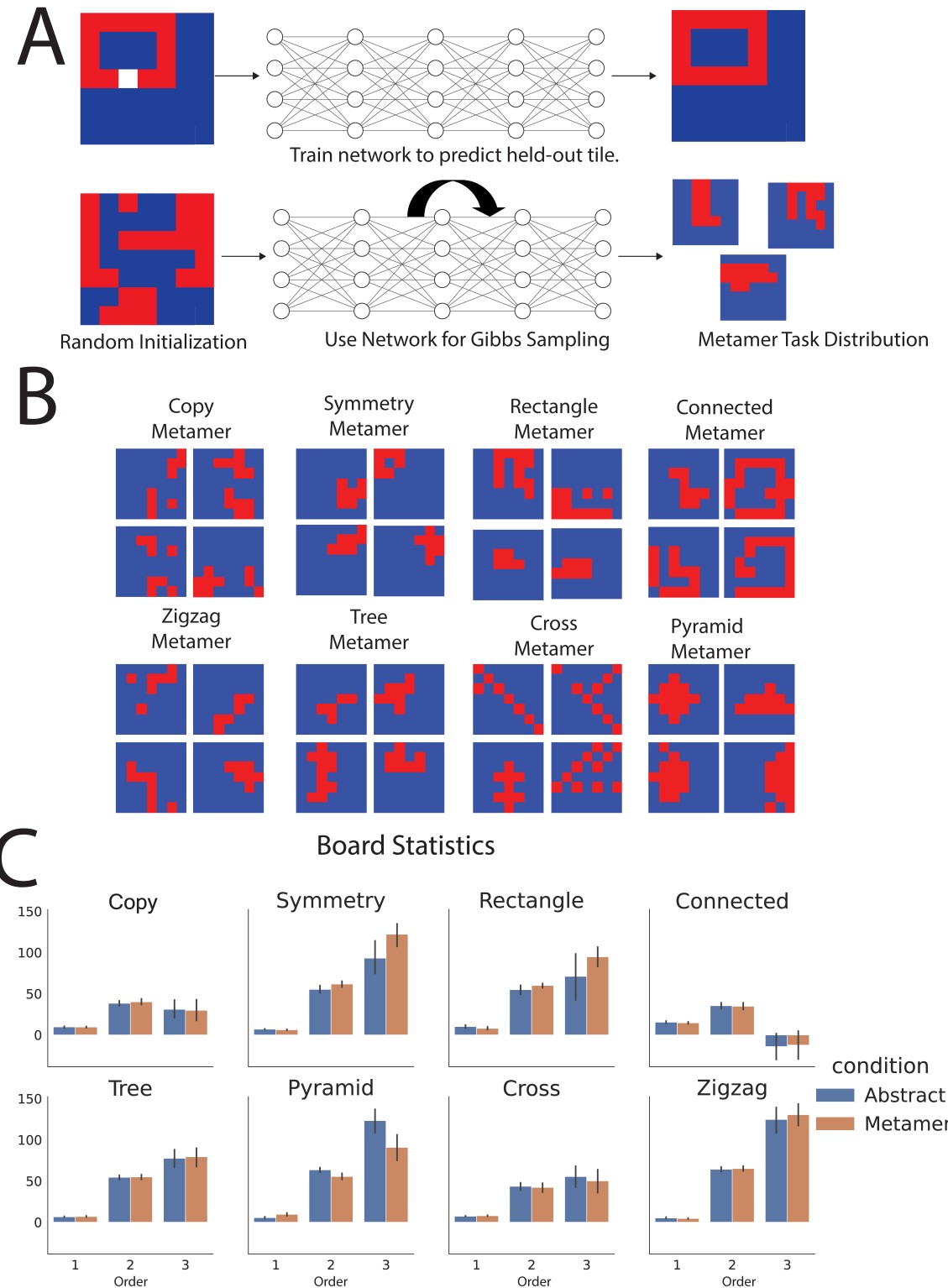

**Fig 2. Metamers for abstract tasks.** (A) Generating metamer distributions. We train a neural network to predict randomly held-out tiles of samples from an abstract rule-based distribution. We then use this network to perform Gibbs sampling. This was done separately for each abstraction. (B) Samples from each abstraction's corresponding metamer distributions. (C) First-, second-, and third-order statistics of boards within each abstract task distribution and their corresponding metamer distribution to show statistical similarity between the distributions. The first-order statistic is the number of red tiles minus the number of blue tiles. The second-order

statistic is the number of matching (i.e. same color) neighbor pairs minus the number of non-matching pairs. The third-order statistic is the number of matching "triples" (i.e. a tile, its neighbor, and its neighbor's neighbor) minus the number of non-matching triples. The statistics were not significantly different across each of the three levels between the abstract and metamer distributions with only two exceptions in the second- and third-order statistics of the pyramid distribution and symmetry distributions.

objective metrics to illustrate the similarity between the abstract and metamer stimuli as a sanity check. **Fig 2B** shows examples of metamer samples from each abstraction. **S3 Fig** shows example sweeps over time of the Gibbs sampling procedure.

It is important to note that, although the main task we study (the tile revealing task, see **Fig 1A**) also involves uncovering tiles conditioned on partially revealed boards, the Gibbs sampling procedure to construct a metamer board distribution and the tile-revealing task are independent. The former is a methodology to procedurally generate "levels" of the game and the latter is the game itself.

## Human versus machine performance on different abstractions

Meta-learning is an established approach for training agents to perform well on task distributions that share common structure. As a first comparison, we used a common recurrent neural network architecture [26,42], trained using Advantage-Actor Critic (A2C) reinforcement learning (see Methods for details). This model has been shown to perform well on task distributions with underlying abstract structure [22,26]. Our question of interest is whether the method described above could be used to differentiate patterns of human and machine performance on abstract versus metamer task distributions

Following the protocol used in previous studies [22,26], we trained this neural network-based meta-learning agent separately on each of our abstract task distributions as well as on each of the corresponding metamer distributions and evaluated performance on tasks held-out during training. We also ran an experiment to evaluate human performance on the same held-out tasks on which we evaluated the agent (see Methods for more details). Note that we did not explicitly train the human participants on the tasks as we did with the agents. We assumed that humans already have inductive biases (innate or learned from prior experience) to perform well on our task, given a simple prompt at the beginning of the experiment that explains that they'll be playing a simple game where they will click on tiles to reveal two-dimensional patterns on a grid to score points.

Results of human and machine performance on the abstraction and metamer distributions are shown in **Fig 3A.** To evaluate performance, we counted the number of blue tiles that humans and agents revealed in the episode (lower is better). To control for the number of red tiles in individual boards as well as general board difficulty, we normalized the performance scores by a "nearest-neighbor heuristic." Specifically, we ran a policy that randomly clicked on unrevealed tiles adjacent to revealed red tiles for 1000 trials for each board, and z-scored the human/agent's performance on the distribution of nearest neighbor heuristic scores. The lower this z-score, the better the human or agent did relative to the nearest neighbor heuristic.

To evaluate the extent to which humans and machines performed differently on the two different types of tasks, we carried out a three-way ANOVA, with performer (human or agent), task type (abstract or metamer), and abstraction (copy, symmetry, rectangle, connected, tree, pyramid, zigzag, and cross) as the factors (see **S2 Table**). We found all effects to be statistically significant and, in particular, that humans and agents differed in their pattern of performance for the abstract versus metamer conditions. The latter was supported by a statistically significant two-way interaction of performer by condition ($F_{1,1036} = 888.716$, $p < 0.001$). As can be seen in **Fig 3A** this effect is driven by a tendency, across tasks, for humans to perform better

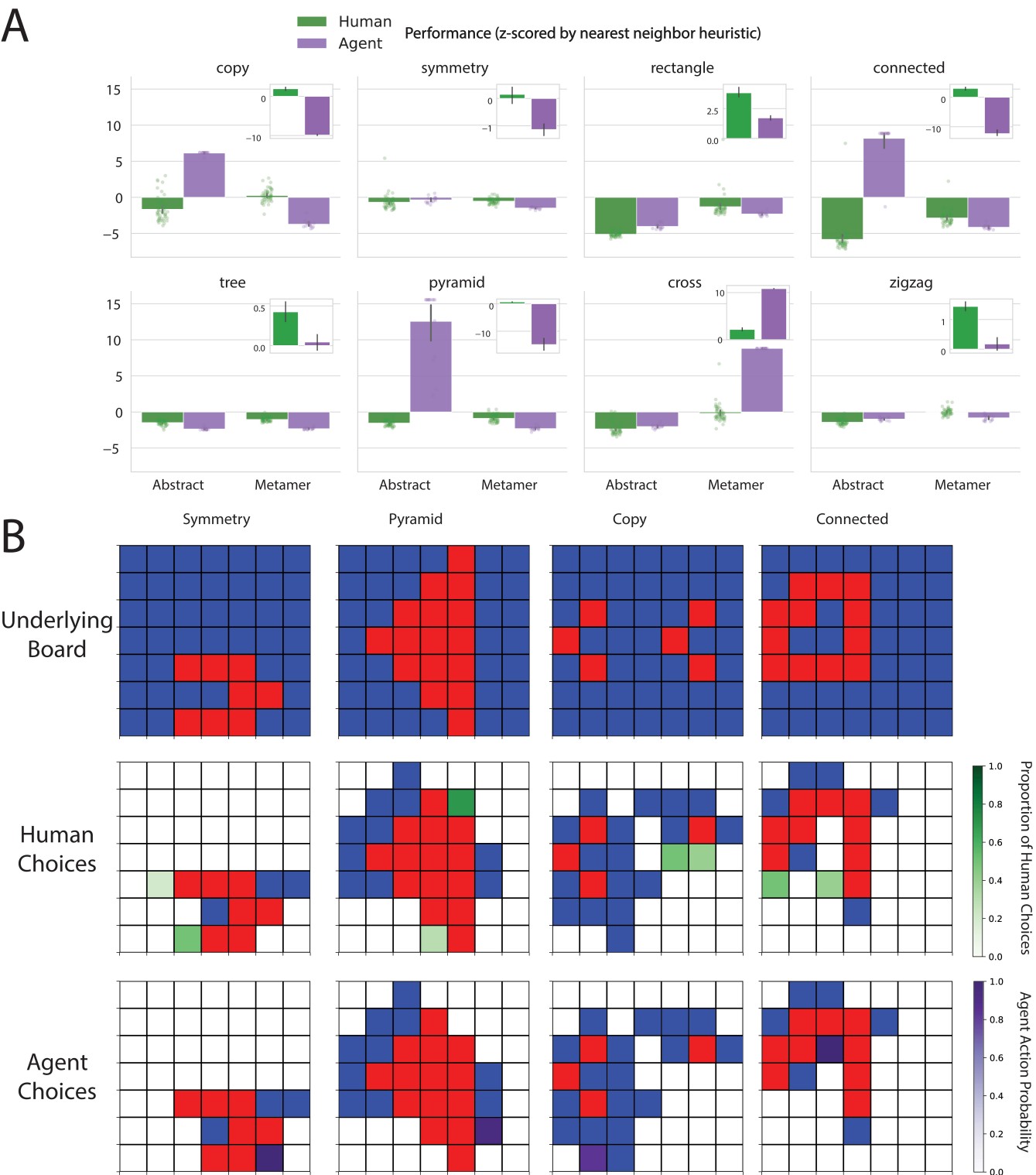

**Fig 3.** (A). Performance of human and neural network agents across all abstractions and their metamers in the tile revealing task. Each dot represents an individual human/agent's mean across tasks in the corresponding task distribution and error bars represent 95% confidence intervals across humans or agents. Performance is the number of blue tiles revealed z-scored by a nearest neighbor heuristic, so a lower number reflects better performance. Inset plots each show the difference between abstract and metamer performance (higher is better for abstract). The difference between abstract and metamer tasks performance in humans is typically larger than that of the agent. (B) Examples of specific choices for a particular board (upper panels) made by humans (green squares in middle panels) versus agents (purple squares in lower panels) in a subset of the abstractions. In all cases, the agent chooses an action (with high confidence) that violates the rule whereas the majority of humans in each case choose the action consistent with the rule.

on the abstract than the metamer tasks, with this tendency attenuated or reversed for agents depending on the task.

Because this effect also differed across abstractions (as shown by a statistically significant three-way interaction, $F_{7,1008} = 339.574515$, $p<0.001$), we examined the abstractions individually using planned comparisons. The performer by task type interaction was statistically significant individually in all eight abstractions (see **S3 Table**). The direction of this effect was also consistent for seven out of eight of the abstractions (all except cross): for those seven, humans performed relatively better on the abstract tasks compared to agents; that is, the difference between mean abstract performance and mean metamer performance was larger for humans than for agents.

We also carried out planned two-sample independent t-tests (across abstractions) to determine the direction of effects for each performer group. Humans performed statistically significantly better on abstract than metamer tasks ($t_{798} = -13.813$, $p<0.001$). In contrast, agents performed statistically significantly better on metamer tasks than abstract tasks ($t_{238} = -4.890$, $p<0.001$). This result indicates that agents do not benefit from the explicit abstract structure in the way that humans do. T-tests for differences within individual abstractions are reported in **S4 Table**.

The effects reported above concerned relative differences between task conditions within performers (human abstract > human metamer; agent metamer > agent abstract). In addition, we found differences between humans and agents within conditions (human abstract vs agent abstract; human metamer vs agent metamer). Humans performed significantly better on abstract tasks than agents ($t_{518} = -13.177$, $p<0.001$). This effect was seen for seven out of eight abstract tasks (all except cross, see **S4 Table**). When aggregating across all eight abstractions, agents did numerically but not significantly better than humans on metamer tasks ($t_{518} = -0.953$, $p = 0.341$). Agents do however perform (statistically significantly) better than humans on metamer tasks in seven out of the eight abstractions (see **S4 Table**). These results indicate that humans generally perform better than agents on tasks generated from abstract rules, whereas agents tend to perform comparably if not better on tasks generated from statistics that are consequences of such abstract rules.

As noted above, despite the clear general trends in the effects across abstractions, there are also exceptions with respect to individual abstractions. In particular, there was one abstraction in which humans did not differ significantly between abstract and metamer tasks (symmetry). In this case, it may be that symmetry seemed harder for humans to recognize as instantiated in our task paradigm. Thus, there appear to be some abstractions that are harder for humans to recognize and use, at least in the context of the current task. Conversely, although in aggregate agents performed better on metamer tasks, there were two abstractions (rectangle and cross) for which they did significantly better in the abstract tasks than metamer tasks. This suggests that, even though on average the agent used in this work was better at learning statistics rather than abstractions, there are still some abstractions that it was able to learn. These exceptions are not surprising, and support the value of our approach in identifying directions for research on what kinds of abstractions are best recognized by humans and machines, respectively.

To further assess the extent to which the method we present can expose not only overall differences between human and agent behavior, but also subtler effects that may vary across particular instances, we carried out an exploratory analysis of specific choice scenarios (**Fig 3B**). Specifically, we considered the subset of abstractions for which agent performance differences between abstract versus metamer tasks were particularly large. We then looked at specific partially completed boards from this distribution and computed the agent's action probabilities over which tile was revealed next (**Fig 3B,** purple tiles in bottom panels). To compare this to human performance, we conducted a follow-up experiment in which, for each of the

abstractions in the subset, we showed ten participants the same partially completed boards (after they had completed the main task; i.e. 25 boards from the corresponding abstraction) and asked them to play out the rest of the board. We report the proportion of humans that picked each tile (**Fig 3B**, green tiles in middle panels).

In all these cases, the agent did not tend to act according to the underlying rule, whereas the majority of humans did. For example, in symmetry, the agent tended to click a tile that is guaranteed to be blue (i.e., violate symmetry). Most humans click on a tile that is guaranteed to be red (consistent with symmetry) and the other humans click on tiles that are not yet guaranteed to be blue (i.e., could still be consistent with symmetry). For pyramid, the agent clicked below the base of the pyramid where there are only blue squares, whereas most humans chose an action to fill out the base of the pyramid. For copy, the agent did not click near where the copy of the shape is supposed to be, whereas all humans did. Additionally, fifty percent of the humans clicked on a tile that is guaranteed to be red due to the copy rule, showing full understanding of the rule. For connected, the agent chose a tile that does not necessarily close the shape whereas all humans chose tiles that will close the currently connected loop. These choice scenarios provide qualitative evidence that humans consistently act in a way that reflects identification of the underlying abstract rules of the task, whereas agents did not show evidence of acting in this way.

To provide further evidence of our insight that the artificial agent often prefers the metamer distribution, we show that, in some cases, the agent that is trained on the rule-based distribution can actually perform better at the metamer boards than held-out tasks from their training rule (**S5 Fig**) This may be a surprising result, because the rule-trained agents never saw any metamers during their training, yet generalize better to the metamer distribution than the distribution they were trained on. We S6 Fig), where training on all eight rules (by randomly sampling from rules during training) yields better test set performance on metamer boards than rule-based boards, despite the metamer boards never having been including during training. We also examined "cross-training" across rules (i.e. training on one rule, testing on another) to examine whether the agent can look for shared structure across rules (S8 Fig).

## Performance of other neural network architectures

To test whether the general pattern of results described above hold for other kinds of neural network architectures, we repeated our experiments for three more architectures (**Figs 4 and S7**): Episodic Planning Networks [43,44], transformers [45], and Compositional Relational Networks [46]. Below is a description of each architecture.

Episodic Planning Networks augment recurrent-based reinforcement learning agents [26] with an external episodic memory implemented through a self-attention mechanism [44]. Our implementation was closely inspired by the implementation of AlKhammasi et al. [43]. It is an augmentation of the LSTM-based meta-learner [26] used in our previous experiment with one additional input. This novel addition is the output of an external memory block implemented as a transformer self-attention layer. The external memory block takes, as input, a fixed number of the previous timesteps' observations and actions and processes it into a self-attention layer [45], whose output is fed into the LSTM which generates an action and value estimate. The network is then trained with A2C, like the model we used in our previous experiments.

Transformers [45] are a class of models that have become increasingly popular in the machine learning literature. We used an implementation inspired by the recent Vision Transformer [47]. The transformer takes in a fixed-size sequence of the individual $7 \times 7$ board's 49 cells (which are one-hot vectors representing covered, uncovered red, or uncovered blue tiles), projects them into a higher dimensional embedding, and processes these embeddings through a sequence of alternating self-attention and feedforward layers. Following the initial ViT work

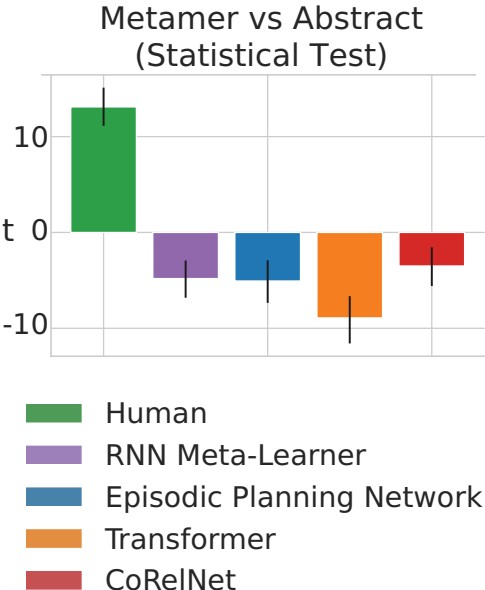

**Fig 4. Results from two-sample independent t-tests that compare metamer vs abstract performance on other neural network architectures.** Humans typically do better on the abstract task distributions (as evidenced by a positive *t* value) whereas the agents do typically worse (as evidenced by a negative *t* value). See **S7 Fig** for performance in human and different neural network architectures across all abstractions and their metamers. Humans typically have a higher difference than the agents, but it is not always the case.

[47] we used learned positional encodings. From the final embedding from the transformer, fully connected layers generate an action and value estimate, and the entire agent is trained with A2C [48].

Compositional Relational Networks (CoRelNet [46]) are an abstraction and simplification of the Emergent Symbols through Binding Network (ESBN [49]). that uses the simplest possible architecture to exhibit an inductive bias towards learning patterns of similarity among inputs in a sequence. Our implementation of CoRelNet takes in the sequence of the 7x7 board's 49 cells (represented as one-hot vectors as we did with the transformer) and computes a $49 \times 49$ similarity matrix among them by taking the dot product among the cells. Then, following Kerg et al. [46], we pass this similarity matrix through a series of fully connected layers. To modify this architecture for reinforcement learning, we outputted an action and value estimate from the fully connected layers, and trained the agent with A2C.

We repeated the same statistical analyses we employed for the RNN Meta-Learner model (three-way ANOVA, see **S5 Table**) in the previous section for each of these alternative neural network architectures. Consistent with our findings for the RNN Meta-Learner model, we found that all of these additional models consistently differed from humans in their pattern of performance across abstract vs metamer tasks (Episodic Planning Networks: $F_{1,1036} = 1176.12352$, $p<0.001$; Transformers: $F_{1,1036} = 2388.503$, p<0.001; CoRelNet: $F_{1,1036} = 712.618$, p<0.001). Furthermore, as with the original RNN Meta-Learning model, we found that each of these models had a significant three-way interaction across performer type (human vs agent), task type (abstract vs metamer), and abstraction (Episodic Planning Networks: $F_{1,1036} = 582.3101$, p<0.001; Transformers: $F_{1,1036} = 242.2132$, p<0.001; CoRelNet: $F_{1,1036} = 127.424$, p<0.001). We repeated follow-up two-way ANOVAs within each abstraction for each of the alternative neural network architectures (see **S6 Table**). Similar to the RNN Meta-Learning

model, where the two-way interaction between human vs agent and abstract versus metamer task was significant in seven out of eight abstractions (all except tree), we found that this effect was significant in eight out of eight abstractions for Episodic Planning Networks, eight out of eight abstractions for Transformers, and seven out of eight abstractions for CoRelNet. We note that the RNN meta-learner typically outperforms the other neural network architectures despite using the same hyperparameter search process (**S7 Fig**; see Methods for more details). This is in line with recent work suggesting that recurrent model-free reinforcement learning models can be very strong baselines for meta-learning tasks [50]. In summary, for all four of the neural network architectures tested, agent behavior differed significantly from human behavior on abstract versus metamer tasks.

As with the RNN Meta-Learner, we also carried out planned two-sample independent t-tests (aggregating across abstractions) to determine the direction of effects for each performer group (see **Fig 4**). We found that, like the RNN Meta-Learner, Episodic Planning Networks ($t_{238}$ = -5.07,$p$<0.001), Transformers ($t_{238}$ = -8.91,$p$<0.001), and CoRelNet ($t_{238}$ = -3.500, $p$<0.001) do significantly better on metamer tasks Results from t-tests on individual abstractions are reported in **S7 Table**.

## Discussion

The ability to recognize and leverage abstract structure is a key aspect of human intelligence [11,12,13,14]. Several lines of research in AI strive to endow artificial systems with this ability. One such approach is meta-learning, in which the intended abstraction is embedded into data, with the intent that training on these data will cause the learner to induce the relevant abstractions. In the work presented here, we show that even when training on data generated by abstract rules, an exemplar deep learning system appears to preferentially encode the statistical structure associated with the generated data rather than the abstract rules from which they were generated. In the past, it has proven difficult to distinguish statistical pattern matching from learning abstractions based solely on test performance of meta-learned agents. Here, we introduce a novel way to disambiguate these using "task metamers." These metamers are generated to have statistical structure that is highly similar to the tasks generated from abstract rules, but that do not use those rules to generate tasks. We instantiate these metamers for a set of abstract rules used to generate tasks of a simple but richly-structured tile-revealing task.

Our results showed that, in contrast to humans who performed significantly better on abstract tasks than corresponding metamer tasks, agents based on common neural network architectures performed comparably or better on the metamer tasks. Additionally, humans generally performed better on abstract tasks than agents whereas agents performed better on metamer tasks than humans. This provides evidence that a range of current deep-learning systems seem to use different strategies than humans. This is consistent with the idea that humans have an inductive bias toward acquiring and using (at least certain forms of) abstract structure, whereas the neural-network learning algorithms tested are generally preferentially sensitive to the statistics that may be downstream consequences of these abstractions. This could be for a variety of reasons, such as the fact that humans have rich prior knowledge making certain tasks easier (e.g. the concept of a rectangle) or the fact that the metamer-generating network could encode statistics in the metamer distribution that makes it easier for the latter artificial agents to learn than for humans.

One important point is that, in many cases, the artificial agent does much more poorly at the rule-based tasks than humans (**Fig 3A**). However, the same architectures, when trained and tested on the respective metamer tasks, often achieves much better performance than humans. Additionally, **S5 Fig** shows that networks trained on abstract tasks can sometimes do better on the out-

of-distribution metamer test set, despite never having been trained on metamers. Both results show us that these networks are not necessarily performing poorly due to being underpowered, but because they differ in inductive bias and encode the "wrong" (i.e. misaligned with the human inductive biases that generated the rule-based stimuli) information when learning these tasks-

It is important to note that, although we found broad trends in the difference in inductive biases between humans and neural-network agents, there were exceptions to these trends. For example, the RNN meta-learner agent we tested did significantly better at some abstract tasks (viz., the rectangle and cross abstractions) compared to their metamer counterparts (see **S2 Table**). Similarly, there were also exceptions in human performance (e.g., performance did not significantly differ between the abstract metamer versions of the symmetry task, see **S3 Table**). In the literature there are similar exceptions. For example, there is evidence of neural networks capable of discovering abstract forms of structure [51], and conversely humans are clearly capable of statistical pattern recognition (e.g. Fiser & Aslin [52]). Recent work has shown that the learning of both rules and statistics in humans have very different learning dynamics, with the former based on a few samples and the latter based on many samples [53]. It will be interesting to see if this is the case for neural networks, which can sometimes learn abstract rules when given lots of rich training experience [51]. We hope that the method we have described for producing and using task metamers to compare performance with explicitly structured tasks will help identify which abstract rules are most easily inferred by humans, as well as which can be learned by neural-network agents.

Our initial focus was on a widely used neural network learning algorithm [26,42]–an LSTM-based meta-learner trained with Advantage Actor Critic (A2C) reinforcement learning [48]–which has previously been used to study whether meta-learning can acquire abstractions [22,26]. Our results suggest that neural network algorithms can be biased to learn statistical structure associated with rules rather than the rules themselves. More importantly, the method we describe provides a means of rigorously evaluating the capabilities of other machine learning algorithms with respect to their ability to infer abstract structure, some of which might be more predisposed to do so than the one we tested. Specifically, we evaluated three other neural network architectures that have been implicated in learning abstractions: Episodic Planning Networks [44], Transformers [45], and CoRelNet [46]. Our results show that, although architectures may vary in their ability to learn abstract structure, many of them do not do so as often as humans do, at least in the tasks we used. However, there may be other architectures (e.g., graph neural networks [54] or neurosymbolic approaches [55]) that are more effective in learning the kinds of abstractions used in our tasks. It may also be possible that a simple architecture can acquire the ability to learn abstraction when trained on large-scale datasets [56]. However, it can be difficult to interpret these systems and quantify the extent to which they've learned such abstractions. Task metamers provide a quantitative approach to examining the inductive biases of such architectures towards structure and their similarity to humans, which can inform the development of human-like machine learning systems.

Since the beginning of work in Artificial Intelligence in the 1950s, many have aspired to build systems that can achieve human-level intelligence. Modern deep-learning systems can often perform tasks that once were thought to be achievable only by humans. Such progress has led not only to technological advances, but also to a refining of our understanding of human intelligence. We hope that the methods and findings we have presented in this article will encourage and facilitate further work seeking to rigorously evaluate and characterize the inductive biases used by humans and machine learning algorithms. With this, we can work towards the twin goal of understanding human cognition and building intelligent systems with the capabilities of human cognition.

## Methods

### Generating metamer task distributions

We trained a fully connected neural network (3 layers, 49 units each) to learn the conditional distribution of each tile given all the other tiles on the abstract rule-based boards. These conditional distributions contain all the relevant statistical information about the boards. We do this by training on an objective similar to those in masked language models like BERT [39] where the goal is to predict a masked-out word in a given sentence. The network was given a board generated with an abstract rule that had a random tile masked out and trained to reproduce the entire board including the randomly masked tile. The loss was the binary cross-entropy between each of the predicted and actual masked tiles, summed over all tiles. The network was trained on samples from the relevant abstraction for 4000 epochs, and typically achieved an accuracy of 95–99%. If the average across 5 epochs was at least 99% accuracy, the training was stopped early.

We used these conditional distributions to generate samples from the distribution of boards learned using Gibbs sampling. We started with a grid in which each tile is randomly set to red or blue with probability 0.5. We then masked out one tile at a time and ran the grid through the network to extract the probability of the missing tile being red or blue from the trained conditional model. We then assign the color of this tile by sampling from this binomial probability. We repeated this by masking each tile in the $7 \times 7$ grid (in a random order) to complete a single Gibbs sweep, and repeated this whole Gibbs sweep 20 times to generate a single sample. The number of sweeps needed for this process to converge was explored by Diaconis et al. [57], where they determined random walks on $n$-dimensional hypercubes can reach its stationary distribution for a number of steps of $N \gg \frac{1}{4} n \log n$. The right hand side is around 20 steps for our case of $n = 49$. Since a sweep is a total of 49 steps (one for each tile), 20 sweeps would be well above this threshold (980 steps). We generate 25 such independent samples from the metamer distribution as held-out test data for the meta-learning agent and sample from this distribution during training (while holding out the test set).

### Training Meta-Learning agents on the Tile-revealing task

Following previous work in meta-reinforcement learning [26,42] we use an LSTM meta-learner that takes the full board as input, passes it through a convolutional layer along with a fully connected layer, and feeds that, along with the previous action (one-hot representation) and reward (scalar value), to 120 LSTM units. It then outputs a scaler baseline (an estimate of the value function) and a vector with a length of the number of actions (the estimated policy). The agent had 49 possible actions corresponding to choosing a tile (on the $7 \times 7$ board) to reveal. The actions are chosen using the softmax distribution of this vector. The reward function was: +1 for revealing red tiles, -1 for blue tiles, +10 for the last red tile, and -2 for choosing an already revealed tile. The agent was trained using Advantage Actor Critic (A2C) (Stable baselines package [58] see [48] for algorithm). We briefly describe the algorithm below.

The loss function of the agent $\mathcal{L} = \mathcal{L}_\pi + \mathcal{L}_v + \mathcal{L}_{ent}$ is a weighted sum of the policy, value, and entropy loss terms respectively (where the weights of each of the value and entropy losses are chosen as hyperparameters). The policy gradient loss term $\mathcal{L}_\pi = \hat{\mathbb{E}}_t[\log \pi_\theta(a_t|s_t)\hat{A}_t]$ directly optimizes the expected advantage of the policy's actions. The term $\log \pi_\theta(a_t|s_t)$ gives the log-probability of a particular action (that is being taken the expectation over) and the term $\hat{A}_t = R_t - V^\pi(s_t)$ is an estimate of the relative value of an action (note that, as mentioned before, $V^\pi$ and $\pi_\theta$ are both outputs of the network). The resulting gradient of this loss function (often referred to as the *policy gradient*) is an unbiased estimator of the expected reward of the policy. The value loss term $\mathcal{L}_v = [V^\pi(s_t) - R_t]^2$ is the mean-squared error between the network's

value estimate and the actual reward, which serves to improve the quality of the agent's estimate of the state's value. The entropy loss term $\mathcal{L}_{ent} = -\mathbb{E}[\pi_\theta(a_t|s_t)\log \pi_\theta(a_t|s_t)]$ is a regularization term equal to the entropy of the current policy and is added to encourage exploration. The parameters of the network are optimized with this loss function through gradient descent and backpropagation through time.

The agent was trained for two million episodes. We performed a hyperparameter sweep (value function loss coefficient, entropy loss coefficient, learning rate, constant/linear learning rate schedule, number of environment steps per update, and discount) to choose the hyperparameters that maximized training reward. See **S5 Table** for the chosen hyperparameters used. The hyperparameter search was done using the Tree-Structured Parzen Estimator [59]. In a single trial, a set of hyperparameters was randomly sampled and evaluated by training the agent for 500,000 episodes and measuring the training reward at the end of 500,000 episodes. Mean training reward was recorded every 50,000 episodes, and a trial was stopped early (i.e. pruned out) if the current trial's mean reward was worse than the median reward of previous trials. For example, if the current trial's mean training reward after 50k episodes was 3 and the median of all previous trials' mean training reward after 50k episodes was 6, then the current trial would be stopped early and a new trial would start. We ran the search for up to 600 trials or up to 80 hours, whichever came first, and chose the best set of hyperparameters across all trials. The selected model was trained for two million episodes and then was evaluated on held-out test grids that were previously unseen. We trained different agents for each abstract task distribution and their corresponding metamer distribution, with separate hyperparameter sweeps for each. **S2 Table** contains the selected hyperparameters for each model and **S4 Fig** contains reward curves for model training.

## Testing Humans on abstract and metamer tasks

We crowdsourced human performance on our task using Prolific (www.prolific.co) for a compensation of $2.25 (averaging ~$13.55 per hour). Participants were shown the $7 \times 7$ grid on their web browser and used mouse-clicks to reveal tiles. Each participant was randomly assigned to one of the eight different abstraction groups (copy, symmetry, rectangle, connected, tree, cross, pyramid, and zigzag) and, within each abstraction, randomly assigned to either the abstract or metamer boards. Each participant was evaluated on the same test set of grids used to evaluate the models (24 grids from their assigned task distribution in randomized order). Note that a key difference between the human participants and model agents was that the humans did not receive direct training on any of the task distributions. Since participants had to reveal all red tiles to move on to the next grid, they were implicitly incentivized to be efficient (clicking as few blue tiles as possible) in order to finish the task quickly. We found that this was adequate to get good performance. A reward structure similar to that given to agents was displayed as the number of points accrued, but did not translate to monetary reward. There were 50 participants in each condition, with eight abstractions and eight metamers. There were sixteen conditions and therefore 800 participants total.

## Alternative neural network architectures

Here we provide descriptions of the neural network architectures alternative to the RNN meta-learner used in this work. Since these models can take longer to converge, each of these models were trained for 8 million episodes (instead of the 2 million used for the original model). We repeated the same hyperparameter search process described in the previous section for each of these model's hyperparameters (see **S8 Table** for final selected hyperparameters for each model).

**Episodic planning networks.**   The implementation of EPNs builds on top of the original LSTM-based meta-learning model used in our initial experiments. Specifically, along with the

CNN-encoded board observation and the previous timestep's action and reward, we add an additional input to the LSTM: the output of an external memory. The input of the external memory is a fixed length memory buffer that keeps track of the sequence of the last $n$ timesteps' boards and actions, where $n$ is the maximum size of the memory buffer and a hyperparameter of the EPN agent. Specifically, for each timestep $i$ in the sequence, a single element of the sequence is a tuple of the previous timestep's observation $o(i-1)$, the action taken at the timestep $a(i)$, and the resulting observation after taking the action $o(i)$. This tuple is meant to represent a single timestep's transition of the underlying MDP of the task (see AlKhammasi et al. 2022)[43]. The sequence of these $n$ tuples is then fed into a single transformer block (self-attention followed by an MLP, see AlKhammasi et al. 2022 and Vaswani et al. 2017)[45] and the output of the transformer block is then fed into the main LSTM of the agent. This LSTM then generates an action and value estimate, and the entire model is trained with A2C (Mnih et al. 2016) [48]. In addition to the standard A2C hyperparameters that we swept over for the original LSTM meta-learning agent, (value function loss coefficient, entropy loss coefficient, learning rate, constant/linear learning rate schedule, number of environment steps per update, and discount), we also swept over: max memory size, dimensionality of embeddings of the transformer block, and number of attention heads.

**Transformers.**   The transformer starts out with a fixed-size sequence of the individual 7x7 board's 49 cells (which are one-hot vectors representing covered, uncovered red, or uncovered blue tiles). This sequence of one-hot vectors is then projected into a higher dimensional embedding space. Following the initial ViT work (Dosovitskiy et al. 2020)[47], we used learned positional encodings. These positional encodings are added to the embeddings in the sequence. Then, the embeddings are processed through a series of alternating self-attention and feedforward layers (see Vaswani et al. 2017 [45] for original implementation of the transformer). An action and value estimate is read out of the output of the transformer, and the whole agent is trained with A2C. In addition to the standard A2C hyperparameters that we swept over for the original LSTM meta-learning agent, (value function loss coefficient, entropy loss coefficient, learning rate, constant/linear learning rate schedule, number of environment steps per update, and discount), we also swept over: number of transformer blocks (one block includes a self-attention and a feedforward layer), dimensionality of embeddings, and number of attention heads.

**CoRelNet.**   Our implementation of CoRelNet takes in the sequence of the $7 \times 7$ board's 49 cells (represented as one-hot vectors as we did with the transformer) and computes a $49 \times 49$ similarity matrix among them by taking the dot product among the board cells. Then, following Kerg et al. 2022[46], we pass this similarity matrix through two fully connected layers. To modify this architecture for reinforcement learning, we outputted an action and value estimate from the fully connected layers, and trained the agent with A2C. We swept over the standard A2C hyperparameters that we swept over for the original LSTM meta-learning agent, (value function loss coefficient, entropy loss coefficient, learning rate, constant/linear learning rate schedule, number of environment steps per update, and discount).

## Supporting information

**S1 Fig. More samples from all abstract task distributions and corresponding metamer task distributions.**
(PDF)

**S2 Fig. Training accuracies across time of the network we use to produce metamers (see Fig 2A).** We trained one network for each abstraction. The network was trained for 4000 epochs or until the average accuracy was above 99%, whichever was first.
(PDF)

**S3 Fig. Example sweeps from the Gibbs sampling process to produce a metamer for the pyramid abstraction.** We start with a random initialization and iterate through the whole board, flipping each tile with the network's given probability. Eleven example sweeps are shown here.
(PDF)

**S4 Fig. Reward curves over episodes for all models trained.**
(PDF)

**S5 Fig. As another means of assessing what the neural network agents learned, we examined how agents trained on *abstract task* distributions perform when given test tasks from the *metamer* distributions.** We do this on the task distributions in which we saw the largest difference between agents and humans. Note that this is an out-of-distribution test and we would normally expect that an agent would have better test performance on tasks that are from the distribution on which it was trained. We find, however, that agents performed similarly or even better on the metamer test tasks than on held-out abstract boards for some of the abstractions. This suggests that agents did not respond to the metamer boards as out of its training distribution. Rather, the agent's behavior indicates that the metamer distribution actually shares the structure it learned during training on the abstract tasks. This is consistent with the hypothesis that the agent learns statistical features *even when directly trained on abstract task distributions*, learning the statistics associated with those abstractions rather than the abstractions themselves
(PDF)

**S6 Fig. In our main results, we train the meta-rl agent on each rule one at a time and test them on held-out examples from that rule.** In this analysis, we trained a single agent on a mixture of all eight rules (by randomly sampling from each one of the rules during training) and evaluated it on held-out examples from each of the eight rules and their metamers. The agent consistently did the same or better on the metamer boards than the rule-based boards in seven out of the eight rules. Note that this agent did not see any metamers during training, yet often generalizes better to the metamer distribution.
(PDF)

**S7 Fig.** (A) Raw performance across all abstract and metamer tasks for humans and all neural network architectures. Inset plots each contain the difference between abstract and metamer performance. (B) Results from two-sample independent t-tests that compare metamer vs abstract performance on other neural network architectures. This is the same as **Fig 4** reproduced here for convenience.
(PDF)

**S8 Fig. Heatmaps of performance on the artificial agents when trained/tested on specific rules.** Each cell denotes the z-score performance (with respect to the nearest neighbor heuristic, as in **Fig 3A**) of the agent when trained on the respective rule of the row and tested on the respective rule on the column. One interesting finding from this map is that all of the training rules produce comparable performance on the copy task, which may suggest the difficulty in learning the unique structure that the copy task employs. Another interesting finding is that training on zigzag metamers leads to good performance on all other metamer tasks, suggesting that the statistics within the zigzag metamers helps the network learn all the other metamer tasks.
(PDF)

**S1 Table. Each abstract task distribution used in this work and their descriptions.**
(PDF)

**S2 Table. Three-way ANOVA analysis of results in Fig 3A.**
(PDF)

**S3 Table. F-values for two-way interactions between human vs agent factor and metamer vs abstract factor within each abstraction for results in Fig 3A.**
(PDF)

**S4 Table. *t* and *p*-values for comparing: mean human performance in abstract vs metamer tasks, mean agent performance in metamer vs abstract tasks, mean human performance vs mean agent performance in abstract tasks, and mean agent performance vs mean human performance in metamer tasks.**
(PDF)

**S5 Table. Three-way ANOVA analysis repeated for alternative neural network architectures (same as S2 Table, which uses the RNN Meta-Learner as the agent for comparison).**
(PDF)

**S6 Table. F-values for two-way interactions between human vs agent factor and metamer vs abstract factor within each abstraction for each alternative neural network architecture (same as S3 Table, but for different architectures).**
(PDF)

**S7 Table. *t* and *p*-values for comparing: mean agent performance in metamer vs abstract tasks for alternative neural network architectures.**
(PDF)

**S8 Table. Selected hyperparameters for each model used.**
(PDF)

## Acknowledgments

We thank Ilia Sucholutsky, Erin Grant, and Jane Wang for helpful feedback on initial versions of the manuscript.

## Author Contributions

**Conceptualization:** Sreejan Kumar, Thomas L. Griffiths.

**Data curation:** Sreejan Kumar, Ishita Dasgupta.

**Formal analysis:** Sreejan Kumar, Ishita Dasgupta.

**Funding acquisition:** Jonathan. D. Cohen, Thomas L. Griffiths.

**Investigation:** Sreejan Kumar, Ishita Dasgupta, Nathaniel D. Daw, Jonathan. D. Cohen, Thomas L. Griffiths.

**Methodology:** Sreejan Kumar, Ishita Dasgupta.

**Project administration:** Sreejan Kumar, Ishita Dasgupta, Nathaniel D. Daw, Jonathan. D. Cohen, Thomas L. Griffiths.

**Resources:** Nathaniel D. Daw, Jonathan. D. Cohen, Thomas L. Griffiths.

**Software:** Sreejan Kumar.

**Supervision:** Nathaniel D. Daw, Jonathan. D. Cohen, Thomas L. Griffiths.

**Validation:** Sreejan Kumar.

**Visualization:** Sreejan Kumar, Thomas L. Griffiths.

**Writing – original draft:** Sreejan Kumar, Ishita Dasgupta.

**Writing – review & editing:** Sreejan Kumar, Ishita Dasgupta, Nathaniel D. Daw, Jonathan. D. Cohen, Thomas L. Griffiths.

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
