## [Decision Letter · Decision Letter 0]

4 Apr 2023

Dear Dr Kumar, 

Thank you very much for submitting your manuscript "Disentangling Abstraction from Statistical Pattern Matching in Human and Machine Learning" for consideration at PLOS Computational Biology.

As with all papers reviewed by the journal, your manuscript was reviewed by members of the editorial board and by several independent reviewers. In light of the reviews (below this email), we would like to invite the resubmission of a significantly-revised version that takes into account the reviewers' comments.

As you can see the Reviewers raise many important questions concerning some methodological aspects that have to be addressed in full. The also suggest many control and additional analyses that will be needed to get a better pictures of the results and possible conclusions that can be drawn from them.

Also, please consider showing all data points using scatter plots or raincloud plots, rather than bar plots with error bars, which hide the distribution of the underlying data.

We cannot make any decision about publication until we have seen the revised manuscript and your response to the reviewers' comments. Your revised manuscript is also likely to be sent to reviewers for further evaluation.

Sincerely,

Stefano Palminteri

Academic Editor

PLOS Computational Biology

Daniele Marinazzo

Section Editor

PLOS Computational Biology

As you can see the Reviewers raise many important questions concerning some methodological aspects that have to be addressed in full. The also suggest many control and additional analyses that will be needed to get a better pictures of the results and possible conclusions that can be drawn from them.

Reviewer's Responses to Questions

**Comments to the Authors:**

Reviewer #1: Review of Disentangling Abstraction from Statistical Pattern Matching in Human

and Machine Learning by Kumar et al

This is a clever study which explores how humans and neural networks learn to perform an RL task reminiscent of Battleships. The task involves predicting the location of hidden tiles on a grid; the tiles obey canonical (researcher-generated) placement rules, for example that they form continuous contour that traces the edge of rectangle, or they cluster together in islands that are symmetrical. Whilst these researcher-generated rules have human-recognisable rule-based semantics, tile configurations that obey each of these rules can also be considered to be samples from a generative model that builds tile grids based on patterns of 1st, 2nd and 3rd order spatial co-occurrence (although the true rules are candidate samples from this model, not all samples from this model will precisely obey the researcher-defined rules). A generative model that encodes these patterns of co-occurrence can thus be used to generate task “metamers”, allowing the researchers to whether neural networks (for example, meta-trained deep networks attempting to find the battleship in a minimal number of moves) really encode the rule-based semantics or whether they rely on the spatial statistics. As the authors put it: “We hypothesize that even on distributions of tasks generated from abstract rules, neural-network meta-learners do not necessarily internalize the abstract structure of those rules, despite performing well. Rather, they may learn the (potentially more complex) statistical structure of the stimuli and corresponding responses that are associated with the rules, without encoding the abstract rules themselves. This is in contrast with humans, who are posited (innately and/or through lifelong learning) to represent, identify, and use abstract rules in such settings (Lake et al. 2015; Lake & Piantadosi 2020; Johnson et al. 2021)”

The authors tackle this by training an additional neural network to behave as the encoder and then sampling successively from this network to produce task metamers, i.e. boards for which the conditional probability of each tile matched that predicted by the encoder. They then verified that there was indeed a good match up to the 3rd order. They then tested humans and (individually) trained meta-RL agents on both rule-based (“abstract”) and metamer boards, showing that in most cases, humans do fairly well on both, whereas the networks make significant numbers of errors on the “abstract” stimuli, which the authors interpret as implying that the networks have encoded the statistics but not the rules.

I think this study will be widely read and cited because it very elegantly makes a simple (and perhaps under-appreciated) point: that where stimuli are generated by hard-and-fast rules, learning the feature statistics can give the misleading impression that the task has been mastered. This is just as true in object recognition (e.g. Jagadeesh 2020) as it is in the current task. Of course, given that the dependencies in the rule-based tasks are special cases of those sampled from the conditional distribution of spatial statistics, the fact you can’t go backwards from the latter to the former is perhaps unsurprising in hindsight. But I think the study is valuable, nonetheless.

I did have few questions about the data that other readers might share, and a few suggestions that go beyond the current report (which I thought interesting but are not essential for the paper)

1/ performance data are normalised using a control model that randomly samples nearby tiles. I wondered if this partly distorted the pattern of naïve rewards obtained in each condition. Surely, if the spatial statistics of the metamers and rule-based tasks are strictly matched, and the claim is that the networks are simply learning the conditional distribution (and not the abstract rules), then they should do equally well on metamers and abstract tasks (whereas humans should do better on the latter because they share priors with the authors, who made the tasks?). But in fig 3A we see that networks do vastly worse on rule-based tasks. Or is this because it is simply harder to generalise these tasks because the training distribution is more narrow?

2/ One analysis which the authors do not run (but which might shed light on this issue) is to compare the performance of networks meta-trained exclusively on metamer distributions but evaluated separately on rules and metamers. Presumably here the strong prediction from the authors’ claim [that the networks are just learning the statistics and not the rules] is that performance would not differ?

3/ Also the networks trained on abstract tasks really just do very badly – z > 5 here implies that every one of 10^5 games was performed worse than the nearest-neighbour random rule. It might be worth explaining why this is? In fact networks do so badly that it’s actually quite difficult to see that people are doing better on abstract rules, which is what you might expect.

4/ we don’t know what the human prior is but presumably it is neither fully based on rules or statistics but a mixture of both. I wonder if the authors could have used different admixtures of rule-based and metamer stimuli in the meta-training set to try to infer this prior? Presumably networks meta-trained exclusively on rule-based stimuli will fail badly on the metamer task (as they largely do on the abstract task) but those meta-trained on just metamers have no reason to do better on the abstract task. But presumably, there is a sampling bias in the human experience of rules that tilts them towards doing better in the abstract task: could the authors re-create this using meta-training (this would be more similar to the Lake work the authors cite).

5/ Finally, does cross-training on different rules help? E.g. does network performance vary on the symmetry task if they are meta-trained on other rules / metamers in parallel? The other rules may contain information that is relevant (e.g. that rule-based patterns tend to obey Gestalt properties such as good continuity) that the network can use at evaluation (conditioning in the red tiles it has revealed so far). Presumably this is more similar to the task that humans face, given that they do not know which of the many possible rules they have encountered in the past might be relevant in the test phase.

Reviewer #2: Stimuli that are generated based on some rule are inherently biased in terms of statistical properties. Successful generalization to held-out stimuli is therefore possible, to some extent, simply based on statistical learning even in the absence of any abstract reasoning (identification of the rule). This concern is particularly strong with AI agents that are trained on a very large number of samples and are known to excel at statistical learning. Here, the authors propose to use metamers, i.e. stimuli that are matched with rule-based stimuli in terms of statistics, and that would therefore be treated in just the same way as rule-based stimuli by an agent that relies on statistics (rather than the identification of the rule). The design of metamers stimuli that are meaningful to that purpose is a key element of this enterprise. The authors present a memater generation strategy in which the statistics that characterize rule-based stimuli are estimated by training a network. Human behavior was markedly different between rule-based stimuli and those statistically matched metamers. By contrast, the behavior of AI agents (different network architectures are tested) was much more similar. This difference indicates that human subjects have internalized (most of) the rules, and that AI agent have internalized only their statistical properties.

The question of what drives the ability to generalize is fascinating. The authors make an interesting methodological proposal. My main concerns are about the specific ways those metamers are created and used. The behavior of human and simulated agents is very interesting, and indicates that they rely on different processes (beyond my concerns about the way metamers were designed). Here are my main concerns.

1) The metamer task is not needed to distinguish between humans and AI agents here: there are clear differences in their behavior. The text sometimes suggests that it is difficult to conclude that the AI agents have not internalized the rule when tested on held-out samples generated by the rule (e.g. p. 6-7: “… examining average performance on held-out examples from structured task distributions. This approach may not reliability distinguish whether a system has truly internalized the underlying abstraction or whether it has learned statistical patterns”). However, during test, it is clear that the AI agents have not internalized the rule, in contrast to humans (e.g. the beautiful examples of Fig 3B). The metamers are presented as being useful in order to test the hypothesis that AI agents have internalized statistics, but is not the internalization of statistics the only alternative to the internalization of rules? Surely, the AI agents have internalized something, because their behavior show clear evidence for some form of structure. Therefore it seems that rule-based held-out stimuli are enough to conclude that they have internalized statistics. If not true, then the authors should explain what could be the alternative(s).

2) Let’s assume that metamers are needed to prove that some agent has internalized some statistics rather than the rule (but see my point 1), perhaps a more informative use of metamers could be to reveal what type of statistics they have internalized. Different classes of metamers could correspond to different classes of statistics. It would be interesting to identify which statistics make stimuli metamers from the view point of the agent. This is not the approach considered by the authors, but it could extend the present work in very interesting ways.

3) I don’t fully understand why the metamers were designed in the specific way described by the authors. The statistical distribution that is learned (for each rule) is the conditional probability of each tile given full information about the other tiles. I have several questions/remarks.

- First, such a distribution is relevant for the last step of the game, when all tiles but one have been revealed. This is a board configuration that subjects probably almost never see (because many tiles remain blank). The stimuli therefore seem to be matched for statistics that subjects are barely exposed to.

- Second, most conditionals actually do not comply with the rules (i.e. the board configuration on which the probability is conditioned on is impossible given the rule). Therefore, such conditionals could not be learned from actual exposure even by an optimal learner, simply because they are never encountered. I imagine that such “never encountered” conditionals should remain at some prior value, possibly a constant value, in an optimal learner. What do these “never encountered” conditionals look like in the trained network? Do the existence and value of such “never encountered” conditionals bias the metamer generation process in some way?

- Third, I don’t understand why an approach with network training is used in the first place. These conditional probabilities (of board configurations compatible with the rule) can all be evaluated analytically when the rule is given. To illustrate, for the rectangle configuration shown in Fig 2A, the conditional probability of the hidden tile being red is necessarily equal to one. For shapes with more degrees of freedom, these conditionals may not all be 1 or 0 but they can nevertheless be computed (e.g. for the tile at the tip of some cross shape, there may be uncertainty about the length of the branch resulting it a probability that is not 0 or 1). In this approach, the “never encountered” conditional probabilities (i.e. the ones incompatible with the rule) should have some default prior value. It therefore seems possible to run the same Gibbs sampling procedure but with the optimal conditionals under a given rule.

- Last, I am wondering whether 20 sweeps of Gibbs sampling are enough. To illustrate, the example rectangle metamers shown seem really different from a rectangle, in comparison some example connected metamers appear close to the rule. Would the rectangle metamers become more like rectangle with more sweeps? (the same concern of early stopping of the Gibbs sampler is true to other shapes)

- The procedure used seems appropriate to match the low-order statistics (up to order 2, and perhaps even 3, Fig 2C), but is this enough? Why stopping at order 3?

Other remarks:

- my understanding is that a generative process characterizes a set or sequence of observations, but not a task, which is more about some cost function (what the agent is asked to do). Therefore, I find the notion of “task distribution” a bit confusing: the distributions referred are really about stimuli, not tasks.

- the following paper on a potential mental language for geometrical shapes seems relevant: Sable Meyer et al, “A language of thought for the mental representation of geometric shapes”, Cognitive Psychology, 2022

- the following paper on the coexistence of rules and statistics in humans also seem relevant: Maheu et al, “Rational arbitration between statistics and rules in human sequence processing”, Nature Human Behavior, 2022. This paper stress that the inference of rules and statistics have different learning dynamics: it is sometimes possible to infer a rule based on a few samples, whereas learning statistics necessarily require many samples. Learning dynamics could thus be informative about what is internalized: if high performance is achieved after a very limited number of samples, it can hardly be due to statistical learning, but more probably to the inference of a rule.

- Fig 3B. Great figure, but why not keeping the same red and blue hues as in the top panel? I was confused by the change in colors.

**Have the authors made all data and (if applicable) computational code underlying the findings in their manuscript fully available?**

Reviewer #1: Yes

Reviewer #2: Yes

PLOS authors have the option to publish the peer review history of their article (what does this mean?). If published, this will include your full peer review and any attached files.

Reviewer #1: No

Reviewer #2: No
---

## [Decision Letter · Decision Letter 1]

29 Jun 2023

Dear Dr Kumar,

We are pleased to inform you that your manuscript 'Disentangling Abstraction from Statistical Pattern Matching in Human and Machine Learning' has been provisionally accepted for publication in PLOS Computational Biology.

Best regards,

Stefano Palminteri

Academic Editor

PLOS Computational Biology

Daniele Marinazzo

Section Editor

PLOS Computational Biology

Reviewer's Responses to Questions

**Comments to the Authors:**

Reviewer #1: thanks for a responsive review. I don't have any further comments.

Reviewer #2: I thank the authors for their responses, the additional analyses they conducted and the changes in the paper.

**Have the authors made all data and (if applicable) computational code underlying the findings in their manuscript fully available?**

Reviewer #1: Yes

Reviewer #2: Yes

PLOS authors have the option to publish the peer review history of their article (what does this mean?). If published, this will include your full peer review and any attached files.

Reviewer #1: No

Reviewer #2: **Yes: **Florent Meyniel

---

## [Editor Report · Acceptance letter]

4 Aug 2023

PCOMPBIOL-D-23-00297R1 

Disentangling Abstraction from Statistical Pattern Matching in Human and Machine Learning

Dear Dr Kumar,

I am pleased to inform you that your manuscript has been formally accepted for publication in PLOS Computational Biology. Your manuscript is now with our production department and you will be notified of the publication date in due course.

With kind regards,

Zsofi Zombor
